# Low Iron Diet Improves Clinical Arthritis in the Mouse Model of Collagen-Induced Arthritis

**DOI:** 10.3390/cells13211792

**Published:** 2024-10-29

**Authors:** Godehard A. Scholz, Sisi Xie, Tasneem Arsiwala, Daniel Guggisberg, Monique Vogel, Martin Bachmann, Burkhard Möller

**Affiliations:** Department for Rheumatology and Immunology, Inselspital, University Hospital Bern, 3010 Bern, Switzerland

**Keywords:** inflammation, iron, TIMP-1, ferroptosis, collagen-induced arthritis, clinical arthritis score

## Abstract

**Background:** In response to inflammation, the absorption of nutritional iron is restricted. Since the pathophysiological significance of the presence and uptake of iron in chronic inflammation is still unknown, we tested the effect of a low iron diet on the clinical course of arthritis in the mouse model of collagen-induced arthritis (CIA). **Methods:** Six- to eight-week-old male DBA/1 mice were fed either a normal (51 mg/kg) or a low iron diet (5 mg/kg) starting four weeks before the first immunization. From day 4 after the second collagen booster made on day 25, the development of arthritis was regularly monitored until the end of the experiment (day 34), using a standard clinical arthritis score. Concentrations of mouse anti-bovine and anti-mouse collagen type 2 IgG antibodies were measured by ELISA; blood cell counts were performed and mediators of inflammation, tissue matrix degradation, oxygenation and oxidative stress were measured in the mouse sera of both diet groups at the end of the experiment by bead-based multiplex assay. Fe^2+^, Fe^3+^, oxidized and reduced glutathione (GSH and GSSG) and malondialdehyde (MDA) were quantified in whole paw tissue by ELISA. Quantitative PCR was performed in the tissues for glutathione peroxidase 4 and other key regulator genes of iron metabolism and ferroptosis. We used nonparametric tests to compare cross-sectional data. Nonlinear regression models were used for longitudinal data of the arthritis scores. **Results:** Mice fed a low iron diet showed a significantly less severe course of arthritis compared to mice fed a normal iron diet (*p* < 0.001). The immune response against bovine and mouse type 2 collagen did not differ between the two diet groups. Mice fed a low iron diet exhibited significantly lower serum levels of tissue inhibitor of metalloproteinase-1 (TIMP-1), a central regulator of inflammation and tissue matrix degradation (*p* < 0.05). In addition, a low iron diet led to a significant reduction in red blood cell indices, indicating restricted iron uptake and latent iron deficiency, but had no effect on hemoglobin concentrations or red blood cell counts. There were no differences between the dietary groups in Fe^2+^ or Fe^3+^ content in the paws. Based on calculation of the GSH/GSSG ratio and high MDA levels, high oxidative stress and lipid peroxidation were likewise detected in the paws of both diet groups of mice. Consequently, no differences associated with gene expression of key regulators of iron metabolism and ferroptosis could be detected between the paws of both diet groups. **Conclusions:** Restricted dietary iron intake alleviates immune-mediated inflammation in CIA without causing anemia. This finding suggests a promising option for dietary treatment of arthritis in inflammation. The underlying mechanism causing reduced arthritis may be linked to the complex regulatory network of TIMP-1 and appears to be independent from the local iron levels, oxidative stress and ferroptosis in the synovial tissues.

## 1. Introduction

Synovial inflammation is one of the hallmarks of rheumatoid arthritis (RA) [1]. Abundant production of pro-inflammatory cytokines and chemokines triggers the invasion of macrophages and lymphocytes into the joint, followed by proliferation of synovial fibroblasts, formation of osteoclasts, tissue matrix degradation and excessive angiogenesis in response to hypoxia and oxidative stress [2]. Since crucial for oxygen transport in health but essential for the proliferation of microbes during infection, iron homeostasis is tightly regulated [3]. Free Fe^3+^ from the blood is transported into the cell by transferrin (TF). When interacting with the transferrin receptor (TFR), receptor-bound iron is taken up into endosomes, where Fe^3+^ is reduced to Fe^2+^ before entering the cytoplasm. Intracellular Fe^2+^ is stored bound to ferritin, formed by ferritin light and heavy chain 1 (FTH1) and, to a lesser extent, as a labile iron pool. During inflammation, intestinal iron availability and the amount of circulating iron are limited due to identical mechanisms on different cell types [4]. Ferroportin (FPN), the only known cellular iron exporter, is expressed in macrophages, duodenal enterocytes and hepatocytes. FPN is a target for the liver-expressed hormone hepcidin, which promotes its internalization and degradation, thereby inhibiting the cellular iron efflux [5,6]. Hepcidin transcription is activated by pro-inflammatory cytokines, most prominently interleukin (IL-) 6, via the signal transducer and activator of transcription (STAT) 3 pathway [7,8]. An alternative mechanism for inhibiting FPN-mediated iron export is its transcriptional repression by pro-inflammatory cytokines [9]. In addition, in the hypoxic inflammatory state, synovial fibroblasts have been shown to produce stroma cell-derived factor 1 (SDF1) and vascular endothelial growth factor (VEGF), the latter being essential for neovascularization [10]. As a consequence, synovial inflammation in RA is characterized by increased iron storage in the reticuloendothelial system [11] and limited intestinal iron absorption [12], the first leading to a shift and functional iron deficiency and the second to true iron deficiency anemia. Since the impact of these hypoferric states on the underlying disease course remains unknown, it represents a challenge for the clinical practitioner in determining whether and how to treat them.

In 2012, an iron-dependent form of non-apoptotic cell death was discovered and termed ferroptosis [13]. Although the current understanding of ferroptosis is not complete, similar features between ferroptosis and RA have been identified, providing a potential link between iron metabolism, oxidative stress and inflammation [14]. Ferroptosis is characterized by the intracellular accumulation of reactive oxygen species (ROS) arising from the reaction between redox-active iron and lipid peroxides [14], which are generated by the oxidation of membrane phospholipids containing polyunsaturated fatty acids in membrane phospholipids [15]. The heavy chain subunit of SLC3A2 and the light chain subunit of SLC7A11 together form system Xc-, a functional cystine/glutamate antiporter. Intracellular cystine is reduced to cysteine, becomes part of reduced glutathione (GSH) and counteracts under the control of glutathione peroxidase 4 (GPX4) the accumulation of lipid peroxides. In addition, high amounts of ROS can be neutralized by GPX4. Thus, system Xc-, GSH and GPX4 are considered the “classic pathway” of ferroptosis regulation [16]. In addition, a probably context-dependent role of acyl-CoA synthetase long-chain family member 4 (ACSL4) has been described, which is involved in the activation of polyunsaturated fatty acids and their incorporation into membrane phospholipids [17].

To study the effectiveness of blocking iron uptake in inflammation and the additive clinical effect of the limited availability of dietary iron in immune-mediated inflammation and its role for tissue matrix degradation and ferroptosis, we further restricted nutritive iron availability in collagen-induced arthritis (CIA), a well-established mouse model of RA. We assessed the clinical arthritis course and concentrations of mouse anti-bovine and mouse anti-mouse collagen type 2 IgG antibodies and of mediators relevantly involved in inflammation and tissue matrix degradation in the mouse sera. Furthermore, we assessed total blood counts, as well as Fe^2+^ and Fe^3+^ levels, the ratio of GSH and oxidized glutathione (GSSG), as readouts of oxidative stress and the levels of malondialdehyde (MDA), indicating peroxidation of unsaturated lipids in whole paws. We also measured key regulators of iron metabolism and ferroptosis in whole paws by qPCR.

Notably, mice fed a low iron diet experienced significantly less severe arthritis progression, suggesting an anti-inflammatory effect, which could be utilized therapeutically. They showed significantly lower red cell indices, indicating an additive dietary effect to iron restriction only by endogenous regulation in inflammation but without anemia. The anti-inflammatory properties of a low iron diet may be linked to interference with the complex network of tissue inhibitor of metalloproteinase-1 (TIMP-1) and most likely take place independently from iron accumulation and ferroptosis in the inflamed tissues.

## 2. Materials and Methods

### 2.1. Mice

Six- to eight-week-old male DBA/1 mice were purchased from Envigo, Amsterdam, The Netherlands. Upon arrival, mice were randomly assigned to groups of five per cage. Mice were housed under temperature (22 ± 2 °C) and humidity (40 to 60%)-controlled conditions with unlimited access to fresh water and food in a 12/12-h light/dark cycle. 

### 2.2. Food

The standard diet (article number: 343200PXV05, Granovit AG, Kaiseraugst, Switzerland) in the animal facility is referred to as normal iron diet, which contains 51 mg iron per kg.

The low iron diet (catalogue number: D18756, Research Diets Inc., New Brunswick, NJ, USA) contained 5 mg iron per kg, i.e., ten times less than the normal iron diet. Mice were placed on the diet upon arrival four weeks before arthritis induction until the end of the experiment. The usual benchmark for a dietary influence on iron metabolism is about four weeks, at least in humans. Despite better data in mice, the four-week interval before arthritis induction was chosen on the assumption that the dietary effect would be fully achieved by the start of arthritis induction.

### 2.3. Arthritis Induction and Experimental Course

Arthritis was induced with bovine type II collagen (MD Biosciences, Zürich, Switzerland), provided as a frozen solution in 0.05 M glacial acetic acid, which was thawed overnight at 4 °C before use. Followed by emulsification with complete Freund’s adjuvant (MD Biosciences, Zürich, Switzerland), on day 0, isoflurane-narcotized mice received an intra-cutaneous injection of 100 µL emulsion (50 µL per side) just above the tail base. On day 21, mice were boostered by intraperitoneal injection of 200 µL of bovine type II collagen (MD Biosciences, Zürich, Switzerland) and sterile phosphate-buffered saline in a 1:1 mixture. On day 34, isoflurane-narcotized mice were sacrificed by a combination of cervical dislocation and intra-cardiac puncture (Figure 1A).

### 2.4. Clinical Arthritis Score

A standard clinical arthritis score [18] was assessed every day from day 21 onwards. GAS, SX or TA performed arthritis scoring of each paw separately in a blinded manner according to the following standardized system: 0: no clinical signs of arthritis; 1: erythema and swelling confined to the digits; 2: erythema and swelling extended to the digits and pads and 3: erythema and swelling involving digits, pads and wrists/ankles; mice avoid using the affected paw. The scores from each paw were summed, resulting in a maximum achievable score of 12 per mouse. Divergent scores between two assessors were reassessed and discussed until an agreement was found.

### 2.5. Complete Blood Counts

A volume of 200 µL blood was collected by intra-cardiac puncture of isoflurane-narcotized mice on day 34 in EDTA tubes (Sarstedt, Nümbrecht, Germany). Analysis was performed with ProCyte Dx (IDEXX Laboratories Inc., Westbrook, ME, USA).

### 2.6. Measurement of Serum Mouse Anti-Bovine and Mouse Anti-Mouse Type 2 Collagen IgG Antibody Concentrations

For quantification of serum mouse anti-bovine and mouse anti-mouse type 2 collagen IgG antibodies, we used their respective assay kits according to the manufacturer’s protocol in a sample dilution of 1:15,000 (Chondrex, Inc., Woodinville, WA, USA, catalogue numbers: 2032 and 2036). Optical density values were measured at 490 nm with a Thermo Scientific™ Varioskan™ LUX microplate reader (Thermo Fisher Scientific, Waltham, MA, USA). The serum for the measurement was collected at the end of the experiment on day 34.

### 2.7. Measurement of Serum TIMP-1, MMP3, IL-6, IL-1β, TNF-α, IFN-γ and VEGF Concentrations

For quantification of the serum TIMP-1, matrix metalloproteinase 3 (MMP3), IL-6, IL-1β, tumor necrosis factor-α (TNF-α), interferon-γ (IFN-γ) and vascular endothelial growth factor (VEGF) concentrations, we used the magnetic bead-based multiplex assay Mouse Luminex^®^ Discovery Assay according to the manufacturer’s protocol (R&D Systems, Minneapolis, MN, USA, catalogue reference: LXSAMLM). Samples were diluted 1:2. Calculation of the cytokine levels was performed with Bio-Plex Manager software (Bio-Rad, Hercules, CA, USA). The serum for the measurement was collected at the end of the experiment on day 34.

### 2.8. Tissue Homogenization

For determination of the iron, GSH/GSSG and MDA levels, we minced and homogenized whole paws with 250 µL extraction buffer. For real-time qPCR, 20% chloroform was added to each sample. Afterwards, paws were homogenized for three minutes at 25 Hz with a precooled TissueLyser II (Qiagen GmbH, Hilden, Germany).

### 2.9. Determination of Iron Levels

Paw iron levels were quantified by a colorimetric iron assay kit (Abcam, Cambridge, UK, catalogue number: ab83366), according to the manufacturer’s protocol. Absorbance was measured at 593 nm with a BioTek microplate reader (Agilent Technologies, Inc., Santa Clara, CA, USA).

### 2.10. Determination of the GSH/GSSG Levels

For quantification of the GSH/GSSG levels, the GSH/GSSG Detection Assay Kit II (Abcam, Cambridge, UK, catalogue number: ab205811) was used, according to the manufacturer’s protocol. Absorbance was measured at Ex 490 nm and Em 520 nm with a BioTek microplate reader (Agilent Technologies, Inc., Santa Clara, CA, USA).

### 2.11. Determination of the MDA Levels

For quantification of the MDA levels, we used the Lipid Peroxidation (MDA) Assay Kit (Abcam, Cambridge, UK, catalogue number: ab118970). Optical density values were measured at 695 nm with a BioTek microplate reader (Agilent Technologies, Inc., California, Santa Clara, CA, USA).

### 2.12. Real-Time qPCR

Total RNA was extracted with GENEzol™ reagent (Geneaid Biotech Ltd., New Taipei City, Taiwan). One milliliter of the reagent was added to 50 mg of the homogenized paw tissue. Samples were shaken vigorously for 10 s and centrifuged at 4 °C and 16,000× *g* for 15 min to separate the phases. The aqueous phase was transferred, and an equal volume of isopropanol was added. Samples were incubated at room temperature for ten minutes, followed by centrifugation at 4 °C and 16,000× *g* for ten minutes. Pellets of mRNA were washed twice in 70% ethanol, dried and resuspended in nuclease-free water. The mRNA was reversely transcribed into cDNA with the High-Capacity RNA-to-cDNA™ Kit (Thermo Fisher Scientific, Waltham, MA, USA). The mRNA levels were assessed by real-time qPCR, using PowerUp™ SYBR™ Green Master Mix (Thermo Fisher Scientific, Waltham, MA, USA) with the 7500 Fast Real-Time PCR System (Thermo Fisher Scientific, Waltham, MA, USA). Ribosomal protein 29 (RPS29) was used as the internal control. PCR amplification was performed over 40 cycles. All primers (synthesized by Microsynth AG, Balgach, Switzerland) are shown in Appendix A. The results are depicted using the 2(-Delta C(T)) method.

### 2.13. Statistical Analysis

Statistical analysis was performed with either Stata software (version 18) or GraphPad Prism software (version 10.1.2). Graphs were plotted with GraphPad Prism software. Nonlinear regression analysis was used for the longitudinal data of the clinical disease course. For statistical analysis of the mouse anti-bovine and anti-mouse type 2 collagen IgG antibodies, serum mediator concentrations and total blood count parameters, we performed unpaired *t*-tests. *p*-values < 0.05 were regarded as statistically significant.

## 3. Results

### 3.1. Mice Fed a Low Iron Diet Develop Less Severe Arthritis

From days 25 to 34, we clinically assessed the severity of arthritis using the clinical score described above (Figure 1A). Of note, arthritis remained less severe in mice fed a low iron diet compared to mice fed a normal iron diet (Figure 1B), suggesting a protective mechanism. No diet-related differences in weight gain or loss were found (Appendix A).

### 3.2. The Two Diet Groups Do Not Differ in Serum IgG Antibody Concentrations Against Anti-Bovine or Anti-Mouse Type 2 Collagen

Measurement of the serum concentrations of mouse anti-bovine and mouse anti-mouse collagen type 2 IgG antibodies did not show differences between mice fed a low iron diet compared to mice fed a normal iron diet (Figure 2), suggesting another underlying mechanism for the reduced arthritis severity in the low iron diet group than a reduced IgG immune response against type 2 collagen. As the control, the serum antibody concentrations in animals of both diet groups without arthritis induction were below the detection range.

### 3.3. Mice Fed a Low Iron Diet Exhibit Significantly Lower Serum Concentrations of TIMP-1

Mice fed a low iron diet exhibited significantly lower serum concentrations of TIMP-1 (Figure 3A). The two diet groups did not differ in concentrations of IL-6 and VEGF (Figure 3B). As the control, serum concentrations of TIMP-1, IL-6 and VEGF were also measured for both diet groups without arthritis induction in experimental round 2. The median IL-6 serum concentrations were raised at the arthritis induction in both diet groups, but the VEGF serum concentrations were in the same range in both diet groups, irrespective of arthritis induction. The serum concentrations of IL-1β were below the detection limit in 26 out of 30 samples from the normal iron diet group and in 22 out of 28 samples from the low iron diet group (and in all samples from both diet groups without arthritis induction), those of TNF-α in 15 out of 30 samples from the normal iron diet group and in 17 out of 28 samples from the low iron diet group (and in all samples from both diet groups without arthritis induction) and those of IFN-γ in 29 out of 30 samples from the normal iron diet group and in 26 out of 28 samples from the low iron diet group (and in all samples from both diet groups without arthritis induction). The serum concentrations of MMP3 were above the detection limit in 30 out of 30 samples from the normal iron diet group and in 27 out of 28 samples from the low iron diet group (and in all samples from both diet groups without arthritis induction).

### 3.4. Low Iron Diet Does Not Affect Hemoglobin Levels or Red Blood Cell Counts

Discoloration of the feces could be observed throughout the experimental course and indicated low iron levels in mice fed a low iron diet (Figure 4A). On day 34, we measured the hemoglobin levels (g/dL), mean corpuscular volumes (MCVs, fl), mean corpuscular hemoglobin levels (MCH, pg) and red blood cell counts (RBC counts, 10^12^/L). As a low iron diet effect, we found reduced erythrocyte indices: MCV and MCH. Interestingly, the low iron diet had no effect on the hemoglobin levels or RBC counts (Figure 4B).

### 3.5. Articular Iron Accumulation and Ferroptosis Remain Unaffected by a Low Iron Diet

In order to assess the local iron distribution in relation to reduced arthritis severity in mice exposed to a low iron diet, we next set out to detect Fe^2+^ (and Fe^3+^) in the total mouse paws. No differences were found between the normal iron diet group and the low iron diet group in this outcome, nor were any differences in tissue iron content associated with the clinical arthritis score (Appendix A). We then set out to assess key players of iron metabolism and hypoxia response in the total mouse paws by qPCR, namely FTH1, TFRC, FPN1, VEGF and SDF1 (Appendix A). However, we could not identify differences in gene expression between the normal iron diet and low iron diet groups. Next, in order to assess oxidative stress in synovial tissues, we measured the levels of GSH and GSSG in total mouse paws (Appendix A). In this comparison, we found no significant differences between the normal iron diet groups and low iron diet groups; the GSH/GSSG ratio indicated a high oxidative state in the paws of both diet groups, with a maximum GSH/GSSG ratio of 1:5. Finally, the MDA levels as readouts for unsaturated lipid peroxidation were measured in the total mouse paws (Appendix A). Again, we did not find differences in the expression between the normal iron diet groups and low iron diet groups. Finally, we assessed gene expression of master regulators of ferroptosis by qPCR, namely GPX4, ACSL4, SLC7A11 and SLC3A2 (Appendix A). However, we could not identify differences in gene expression between the normal iron diet and low iron diet groups.

## 4. Discussion

By restricting the availability of dietary iron in the highly inflammatory state of the CIA mouse model, we observed a less severe clinical course of arthritis compared to mice fed a normal iron diet. Mice fed a low iron diet exhibited significantly reduced concentrations of TIMP-1 in their sera at the end of the experiment but no significant differences in the primary immune response against bovine collagen type 2, in the secondary immune response against murine collagen type 2 nor the MMP3 or cytokine concentrations in a single measurement at the end of the experiment. Lower erythrocyte indices in mice on the low iron diet indicated a reduction of the available iron for hemoglobin synthesis but without causing anemia or lowering the RBC counts. Importantly, although the tissue iron concentrations in the affected limbs were not significantly altered, we could demonstrate that a well-tolerated low iron diet had an additive and beneficial effect on the physiological regulation of iron uptake in response to inflammation.

Although the CIA mouse model has several differences from human disease [19], it is the standard animal model of arthritis for human RA, closely resembling several of its inflammatory features [20]. To avoid influencing the experimental course by additional blood sampling, blood was collected only at the end of the experiments. Several parameters were quantified in a single dilution in a multiplex assay, which limits the interpretation of the results for parameters that were outside the quantifiable range. For ethical reasons, the number of animals in the experiments was limited to the necessarily acceptable minimum. Another point to consider regarding the experimental setup is that there are two main categories of laboratory animal diets: grain-based diets, such as the normal iron diet, and purified ingredients diets, such as the low iron diet. In contrast to purified ingredients diets, in grain-based diets, the ingredients are unrefined, which might lead to a batch-to-batch variability and difficulties in the replicability of the experiments, though we were not faced with this.

Against this background of limitations, our data have significant translational implications, as the clinical practitioner is often confronted with the interpretation of combined functional and true hypoferric states in inflammatory diseases. While the impact of hypoferric states on the course of the disease remains completely unclear, treatments are nevertheless made to counter-regulate these states [21]. Essentially, the corresponding treatment strategies usually start with oral iron supplementation. However, importantly, oral supplementary strategies for functional iron deficiency remain more or less unsatisfactory, since iron absorption and export require FPN, which is degraded upon hepcidin production in response to inflammation. In addition, although parenteral iron supplementation appears to be safe in the case of stimulated erythropoiesis [22], it was recently shown that at least the intravenous route of iron administration can even aggravate inflammation [23,24]. Accordingly, our data also argue against iron supplementation in inflammatory arthritis but rather suggest a positive role of enteral iron restriction. As such, a hypoferric state in chronic inflammation should be understood as a biomarker of the body’s anti-inflammatory defense, which should not be counter-regulated but therapeutically supported. Of note, positive effects of iron chelator therapies in the management of other chronic inflammatory disorders than arthritis have been described [25]. However, in patients with inflammation-related anemia, the fine-tuning of iron restriction could be quite relevant to not impair quality of life by inducing severe anemia but to take advantage only of its anti-inflammatory effect. Interestingly, in our experiments, a low iron diet led to a reduction in MCV and MCH but was not relevant for the hemoglobin concentrations or RBC counts. Thus, future clinical research will have to assess how such a best balance could look like and whether simple dietary iron restriction could be used as an adjunct to the currently applied biological (b) or targeted synthetic (ts) disease-modifying anti-rheumatic drugs (DMARDs).

Attempting to identify the mechanism of reduced arthritis severity in mice fed a low iron diet, we could rule out an attenuated mouse anti-mouse collagen type 2 IgG response. Acknowledging the difficulties in reliably displaying disease activity and progression by measuring cytokine serum concentrations in CIA [26], we decided to expand on the outcomes covering matrix destruction. Strikingly, we found the serum concentrations of TIMP-1 significantly reduced in the low iron diet group. Although this finding needs further mechanistic confirmation, it nevertheless finds strong support in a report that adenoviral overexpression of TIMP-1 in CIA resulted in a statistically significant increase in inflammation [27]. Of note, the authors described their results as “paradoxical”, owing to their focus only on the metalloproteinase inhibitory activities of TIMP-1, a way of thinking, where TIMP-1 overexpression should lead to a decrease in inflammation. Fortunately, meanwhile, the versatile functionality and mechanistic complexity of TIMP-1 in inflammation was further deciphered. Importantly, the two-domain structure of TIMP-1 exhibits context-dependent, metalloproteinase inhibitory or cytokine-like signaling activities [28]. In line with this, our observation that the MMP3 serum concentrations were above the detection limit in both diet groups with and without arthritis induction and not only in the normal iron diet group with arthritis induction indicates that the arthritis-mitigating effect of the hypoferric state may be mediated by interference with the cytokine-like signaling activities of TIMP-1 and take place independently from matrix metalloproteinase activities. What is more, a link between reduced TIMP-1 serum concentrations, a hypoferric state and attenuated inflammation may exist due to the highly interesting fact that TIMP-1 is identical to the protein named “erythroid potentiating activity” (EPA) [29,30]. EPA is an enhancer of in vitro proliferation of erythroid progenitor cells [31]. This possible connection was completely neglected for almost forty years, but our findings now bring it back into focus.

Due to the common features of iron overload and lipid peroxidation, some evidence from other studies suggests the involvement of ferroptosis in the pathogenesis of RA. However, the current findings are quite controversial, and the question as to whether ferroptosis is protective or harmful in RA remains elusive [32]. In a lipopolysaccharide-induced synovitis model in one report, an increase in iron content and TFRC, as well as a decrease in GPX4, SLC7A11 and SLC3A2, was described, leading to increased cell death [33]. In contrast, other authors have reported decreased ferroptosis associated with decreased ACSL4 levels and increased levels of FTH1, GPX4 and SLCA11 [34]. Further investigating this question, we hypothesized diet-related differences in iron accumulation in the mouse paws, indicating different ferroptotic activity and providing a potential explanation for the different progression of arthritis severity. Somewhat unexpectedly, iron concentrations, oxidative stress and ferroptosis markers were not significantly affected by the low iron diet, thereby either suggesting other mechanisms, which could even take place outside of the affected joints, or too subtle differences to be captured by our methods. In detail, since we did not find differences in the Fe^2+^ or Fe^3+^ content between the paws of mice fed a low iron diet or a normal iron diet, this could suggest that ferroptosis may not be mechanistically involved in the observed anti-inflammatory effect of a low iron diet. However, the effects of ferroptosis may be cell type-dependent, which may also explain the ongoing controversy about its beneficial or harmful effects, and perhaps needs to be deciphered on a single-cell basis in the different tissues of the joint, whereas we can currently only provide global data for all cell types in the inflamed paws. In the context of single-cell analysis, synovial fibroblasts deserve special attention as key players in the pathogenesis of RA, which seem to be selectively regulated via ferroptotic mechanisms in CIA [35]. In addition to our findings regarding the iron content, we could not find any diet-specific differences for FTH1, TFRC, FPN1, VEGF and SDF1 on the gene expression level in the paws. While we were able to detect high oxidative stress levels in the paws of both diet groups based on the GSH/GSSG ratio, there were no group-specific differences or for MDA, the readout for lipid peroxidation. Finally, the gene expression levels of the master regulators of ferroptosis GPX4, ACSL4, SLC7A11 and SLC3A2 were not significantly different between the two diet groups. Although tempting to speculate, all these findings together are arguing against ferroptosis as a general key mechanism of reduced arthritis severity in hypoferric states.

In summary, we report here for the first time that non-hemoglobin- and non-RBC count-relevant oral iron restriction can positively influence the clinical course of arthritis. The induction of such a state could easily be harnessed therapeutically. The mechanism underlying our observation suggests a previously completely unnoticed interaction with TIMP-1 but requires further confirmation.

## Figures and Tables

**Figure 1 cells-13-01792-f001:**
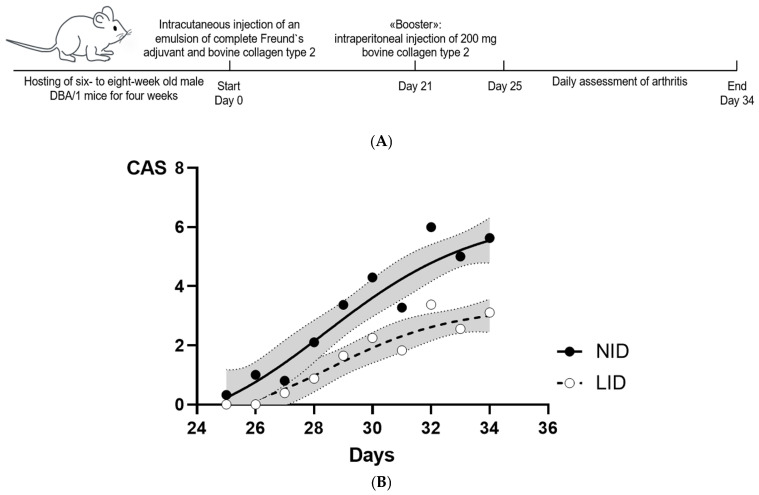
(**A**): Overview of the experimental course. (**B**): Development of arthritis from day 25 to day 34, presented as clinical arthritis score in mice on a normal iron diet or low iron diet. Shown are the mean and (in grey) 95% confidence intervals of the clinical arthritis score (CAS) from six independent experiments, corresponding to biological replicates. Notably, arthritis remained less severe in mice fed a low iron diet in comparison to mice fed a normal iron diet (*p* < 0.0001). NID: normal iron diet; LID: low iron diet.

**Figure 2 cells-13-01792-f002:**
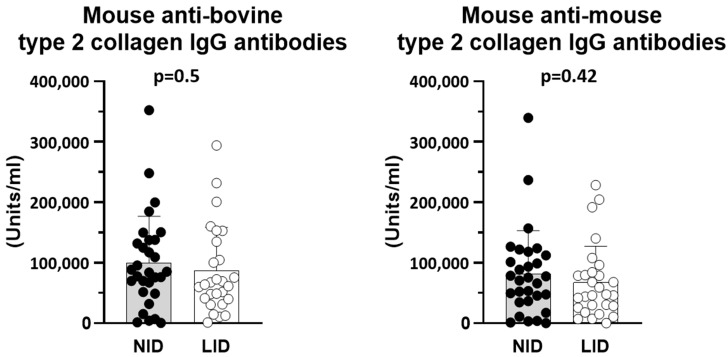
Serum concentrations (Units/mL) of mouse anti-bovine and mouse anti-mouse collagen type 2 IgG antibodies. Equivalent concentrations of mouse anti-bovine type 2 collagen IgG antibodies indicate no differences in the immunological strength of arthritis induction between the two diet groups. NID: normal iron diet; LID: low iron diet.

**Figure 3 cells-13-01792-f003:**
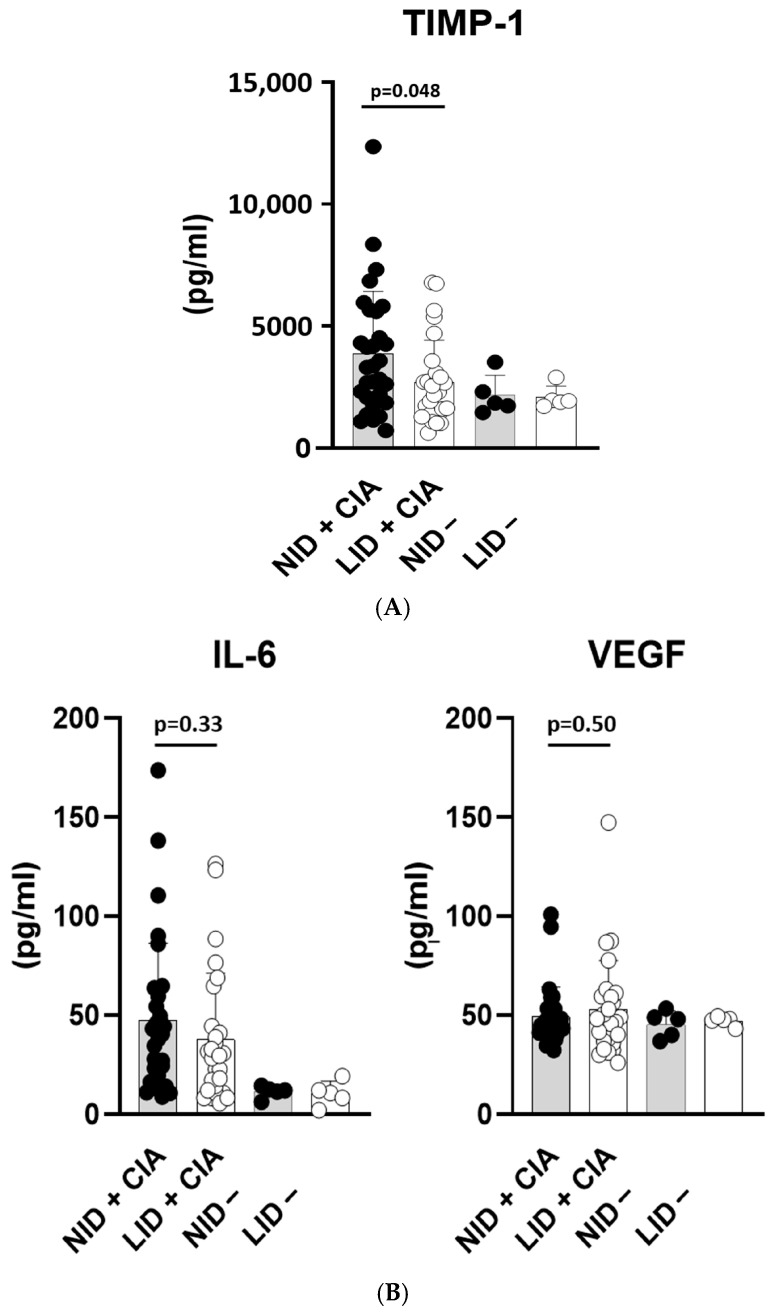
Serum concentrations (pg/mL) of (**A**) TIMP-1 and (**B**) IL-6 and VEGF. Mice fed a low iron diet exhibit significantly lower serum concentrations of TIMP-1 than mice fed a normal iron diet. As the control, the serum concentrations were also measured for both diet groups without arthritis induction. NID: normal iron diet; LID: low iron diet; CIA: collagen-induced arthritis; + CIA: arthritis induced; −: arthritis not induced.

**Figure 4 cells-13-01792-f004:**
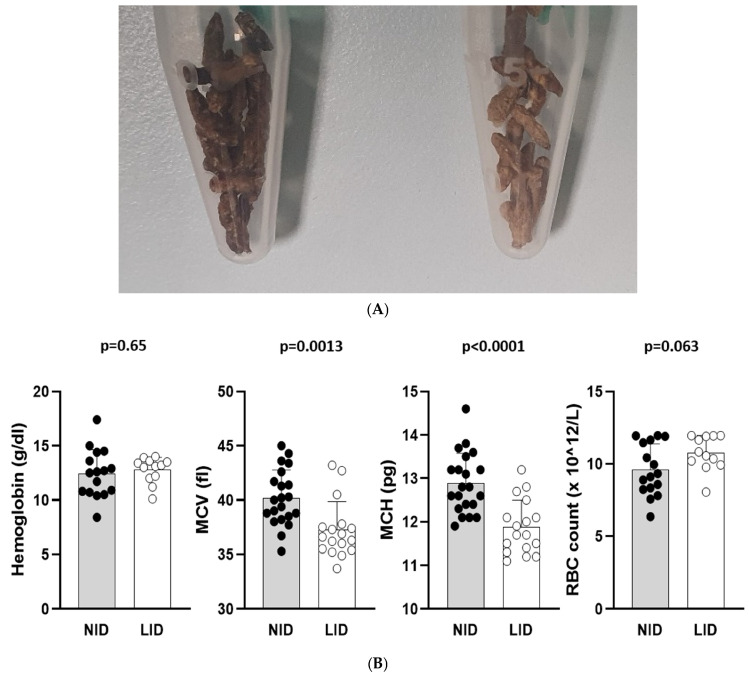
(**A**): Effect of a low iron diet due to discoloration of mouse feces. Left side: normal iron diet; right side: low iron diet. (**B**): Hemoglobin levels (g/dL), mean corpuscular volume (MCV, fl), mean corpuscular hemoglobin (MCH, pg) and red blood cell counts (RBC counts, 10^12^/L) in mice fed a normal iron diet or a low iron diet. Of note, a reduction in MCV and MCH was found in mice fed a low iron diet, although a low iron diet had no effect on the hemoglobin levels or RBC counts. NID: normal iron diet; LID: low iron diet.

## Data Availability

The original contributions presented in the study are included in the article/Appendix A, and further inquiries can be directed to the corresponding authors.

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
