# Peer review of "Low Iron Diet Improves Clinical Arthritis in the Mouse Model of Collagen-Induced Arthritis"

_cells, 2024, doi:10.3390/cells13211792_

Round 1

Reviewer 1 Report

Comments and Suggestions for Authors

This study found that a low-iron diet could alleviate the clinical scores of collagen-induced arthritis. Clinically, patients with rheumatoid arthritis might also suffer from anemia, which sometimes requires iron supplementation. However, iron overload may cause inflammation and potentially worsen arthritis. Therefore, iron supplementation in clinical practice needs to be balanced carefully. This is an interesting study, but it should be improved:

Major Comments:

  1. Given the interesting result that a low iron diet improves arthritis, it would be better to have more comprehensive experiments to validate the findings. For example, serum levels of inflammatory cytokines such as TNF-α, IL-6, and IL-1β, and anti-type II collagen antibody levels if possible. Histological examinations. 
  2. The investigation into potential mechanisms is somewhat superficial. qPCR from whole tissue is not enough to get the conclusion. The speculation and discussion are somehow meaningless. However, as a preliminary study, I think detailed mechanism is not strictly required.

Other Comments

  • The standard diet and low-iron diet are from different companies. Are there other nutritional differences?
  • In 1b, does "six independent repetitions" refer to 6 biological replications or technical replications?
  • In the supplementary tables, what does "round" mean, and why is round 5 missing while there are 6a and 6b? 
  • Why is there missing data in Supplementary Table 3?
  • What about other data like RBC count?
  • Although there are no significant differences, please convert the data in the tables into additional graphs.

Reviewer 2 Report

Comments and Suggestions for Authors

The article is very interesting and well written. It discusses the issue regarding an iron-poor diet and its effect on collagen-induced inflammation in pyr mice. 

My comments are as follows:

1. In Methods, Clinical arthritis score, it says that "clinical arthritis score [17] was assessed every day from day 21 onwards", however, in Supplementary Table 2, the data are presented as Round 1 to 4 and Round 6a and b, while in low iron diet group is Round 2 to 4 and Round 6a and b. The authors need to present more precisely the method of clinical scoring. There is a difference from what is written in the article (namely daily measurement) and what is given in the table in question. On the other hand, in the methods it is said that Score 3 is " erythema and swelling involving digits, pads and wrists/ankles", which is automatically counterbalanced to presented data in Supplementary Table 2 as separate Paw and ankle diameters. This also contradicts the statement that "The scores from each paw were summed".

2. The basic statement that "arthritis remained less severe in mice fed a low iron diet compared to mice fed a normal iron diet" is not substantiated by the results presented. On the other hand, the following sentence negates the previous one, namely "Measuring the ankle and paw diameters of the mice on day 34 revealed no differences between mice fed a normal iron or a low iron diet".  In the cited article No. 17, it is said, "a subjective scoring system is applied to each limb using a scale of 0-4" and an alternative is the use of a plethysmometer. 

3. Figure 1B, is not presentable. It says, "the clinical arthritis score from six independent experiments.", something that does not match the daily measurement described in methods.  In Figure 1B, it is not clear at which point p<0.0001. 

4. Most likely, this sentence should be revised - However, the anti-inflammatory properties of low iron diet take most likely take place independently of iron accumulation and ferroptosis in the inflamed tissues.

5. 100μl, 0.05M, 593nm 50mg and similar - put a space between the numeral and the sign - 100 μl, 0.05 M, 593 nm, 50 mg

Methods and results must be improved.

Comments on the Quality of English Language

Minor editing of English language required

Reviewer 3 Report

Comments and Suggestions for Authors

1) The authors mention that the restricted dietary iron intake alleviates immune-mediated inflammation. However, the author did not evaluate inflammatory parameters, such as inflammatory cytokines. It would be interesting to add experiments evaluating inflammatory markers in mice paws. For example, IL-1β could cause lipid peroxidation and ferroptosis. 

2) What is the rationale for not adding the negative (without disease) and positive (arthritic animals and without diet) control groups? Please, explain this point. 

3) The rationale for feeding animals for 4 weeks before inducing arthritis is unclear. This is not stated in the methods.

4) What is the rationale for selecting at day 34 after disease induction to evaluate gene expression involved in iron metabolism and ferroptosis?

5) Alternatively, this reviewer strongly recommends organizing the gene expression results into a graph.

6) Fibroblast-like synoviocytes are key components in articular inflammation. The authors should discuss the potential involvement of these cells in their findings because their effect on the in vivo results cannot be excluded.  

Round 2

Reviewer 1 Report

Comments and Suggestions for Authors

The authors conducted additional studies and found lower levels of TIMP-1 in the LID group, supporting the hypothesis that LID may alleviate RA by reducing cartilage degradation and destruction. It also suggests a possible mechanism of LID.

However, it is regrettable that other inflammatory factors, such as TNF-α and IL-1β, were not effectively detected. It is difficult to state that the system inflammation was reduced by LID. While LID appears to be associated with lower MDA levels, there was no significant improvement observed in the GSH/GSSG ratio.

In some figures, although not statistical significant ( p>0.05 in Fig. 2, IL-6 and MDA), it would still be helpful to indicate the p-values for reader reference.

Further, the figures must be re-organized, as some of them are displayed with suboptimal quality (Size, position, the figure legend position, ect.)

If the authors wanted to continue their research in the future, I suggest to include histological and radiologic analysis to evaluate arthritis severity. It would be better to give more solid evidence. Additionally, more detailed analysis of systemic inflammation and oxidative stress markers would be beneficial. Detailed mechanistic studies are not essential at this stage, but more in-depth research would certainly be anticipated and appreciated.

Reviewer 2 Report

Comments and Suggestions for Authors

The manuscript investigates the effect of a low-iron diet on immune-mediated inflammation in collagen-induced arthritis. Based on the data, a low-iron diet may have an anti-inflammatory effect. The authors report that non-hemoglobin and non-RBC 483 count-relevant oral iron restriction can positively influence the clinical course of arthritis. 

The manuscript has been revised and improved and is ready for publication. 

Author Response

The manuscript investigates the effect of a low-iron diet on immune-mediated inflammation in collagen-induced arthritis. Based on the data, a low-iron diet may have an anti-inflammatory effect. The authors report that non-hemoglobin and non-RBC count-relevant oral iron restriction can positively influence the clinical course of arthritis. The manuscript has been revised and improved and is ready for publication.

We sincerely thank reviewer 2 for her/his valuable comments and her/his kind recommendation for acceptance of our manuscript for publication.

Reviewer 3 Report

Comments and Suggestions for Authors

The authors have revised the manuscript according to the suggestions provided and satisfactorily addressed the reviewer's inquiries. Based on these, I recommend that the manuscript be accepted. 

Author Response

The authors have revised the manuscript according to the suggestions provided and satisfactorily addressed the reviewer's inquiries. Based on these, I recommend that the manuscript be accepted.

We sincerely thank reviewer 3 for her/his valuable comments and her/his kind recommendation for acceptance of our manuscript for publication.